# Risk Factors for Isolated Sphenoid Sinusitis after Endoscopic Endonasal Transsphenoidal Pituitary Surgery

**DOI:** 10.3390/diagnostics14070758

**Published:** 2024-04-02

**Authors:** Yun-Chen Chang, Yu-Ning Tsao, Chi-Cheng Chuang, Cheng-Yu Li, Ta-Jen Lee, Chia-Hsiang Fu, Kuo-Chen Wei, Chi-Che Huang

**Affiliations:** 1Department of Otolaryngology-Head and Neck Surgery, Linkou Chang Gung Memorial Hospital, No. 5 Fu-Shin Street, Guishan District, Taoyuan City 333, Taiwan; vivian_chang05@hotmail.com (Y.-C.C.); yuning526210@hotmail.com (Y.-N.T.); entlee@cgmh.org.tw (T.-J.L.); fufamily@cgmh.org.tw (C.-H.F.); 2Department of Neurosurgery, Linkou Chang Gung Memorial Hospital, No. 5 Fu-Shin Street, Guishan District, Taoyuan City 333, Taiwanmick791212@hotmail.com (C.-Y.L.); 3Department of Otolaryngology-Head and Neck Surgery, Xiamen Chang Gung Hospital, Xiamen 361028, China; 4Graduate Institute of Clinical Medical Sciences, College of Medicine, Chang Gung University, No. 259, Wenhua 1st Rd., Guishan District, Taoyuan City 333, Taiwan; 5Department of Neurosurgery, New Taipei Municipal Tucheng Hospital (Built and Operated by Chang Gung Medical Foundation), No. 6, Sec. 2, Jincheng Rd., Tucheng Dist., New Taipei City 236, Taiwan; kuochenwei@cgmh.org.tw

**Keywords:** transsphenoidal surgery, endoscopic approach, postoperative sinusitis, pituitary tumors, risk factors

## Abstract

(1) Background: Transsphenoidal pituitary surgery can be conducted via microscopic or endoscopic approaches, and there has been a growing preference for the latter in recent years. However, the occurrence of rare complications such as postoperative sinusitis remains inadequately documented in the existing literature. (2) Methods: To address this gap, we conducted a comprehensive retrospective analysis of medical records spanning from 2018 to 2023, focusing on patients who underwent transsphenoidal surgery for pituitary neuroendocrine tumors (formerly called pituitary adenoma). Our study encompassed detailed evaluations of pituitary function and MRI imaging pre- and postsurgery, supplemented by transnasal endoscopic follow-up assessments at the otolaryngology outpatient department. Risk factors for sinusitis were compared using univariate and multivariate logistic regression analyses. (3) Results: Out of the 203 patients included in our analysis, a subset of 17 individuals developed isolated sphenoid sinusitis within three months postoperation. Further scrutiny of the data revealed significant associations between certain factors and the occurrence of postoperative sphenoid sinusitis. Specifically, the classification of the primary tumor emerged as a notable risk factor, with patients exhibiting nonfunctioning pituitary neuroendocrine tumors with 3.71 times the odds of developing sinusitis compared to other tumor types. Additionally, postoperative cortisol levels demonstrated a significant inverse relationship, with lower cortisol levels correlating with an increased risk of sphenoid sinusitis postsurgery. (4) Conclusions: In conclusion, our findings underscore the importance of considering tumor classification and postoperative cortisol levels as potential predictors of postoperative sinusitis in patients undergoing transsphenoidal endoscopic pituitary surgery. These insights offer valuable guidance for clinicians in identifying at-risk individuals and implementing tailored preventive and management strategies to mitigate the occurrence and impact of sinusitis complications in this patient population.

## 1. Introduction

Pituitary tumors encompass a broad range of intracranial neoplasms, exhibiting a spectrum from benign to malignant. Within this spectrum, patients diagnosed with pituitary neuroendocrine tumors (pit-NETs) often undergo surgery as the primary treatment modality [1]. Transsphenoidal approaches have emerged as the cornerstone of surgical management for most pituitary tumors due to their favorable outcomes, including lower rates of surgical complications and endocrine disturbances. Traditionally, this surgical intervention could be performed using either a microscopic or an endoscopic approach. However, in recent years, the endoscopic technique has garnered increasing favor and has become the mainstream approach for pituitary surgery. This shift can be attributed to several advantages offered by the endoscopic method, including improved visualization of the surgical field, enhanced maneuverability within the narrow confines of the nasal cavity, and reduced morbidity associated with nasal manipulation [2]. As a result, endoscopic transsphenoidal pituitary surgery has become the preferred choice for many surgeons and patients alike, reflecting ongoing advancements in surgical techniques and technology aimed at optimizing patient outcomes and quality of life [3].

While considered a reasonably safe procedure, endoscopic transsphenoidal pituitary surgery has been associated with various complications. Intraoperative complications may include injuries to the optic nerve or extraocular muscles, hemorrhage, and cerebrospinal fluid (CSF) leaks. Postoperative complications tend to be of lesser severity and can include sphenoidotomy stenosis, nasal septal perforation, sinusitis, and the formation of mucocele [4,5]. Among these complications, isolated sphenoid sinusitis following transsphenoidal pituitary surgery is relatively uncommon and has not received comprehensive attention in previous studies. A few retrospective analyses have reported the incidence of sphenoid sinusitis as ranging from 6.2% to 7.5% in patients undergoing endoscopic transsphenoidal surgery [6,7]. In these cases, patients typically needed medical treatment or further endoscopic management. However, until now, there has been a paucity of reports addressing the risk factors for postoperative sphenoid sinusitis. In light of this, our study aims to fill this void by investigating the risk factors associated with the development of isolated sphenoid sinusitis following endoscopic transsphenoidal pituitary surgery. By unraveling these factors, we hope to contribute valuable insights to enhance the understanding and management of this relatively rare complication in the realm of pituitary surgery.

## 2. Materials and Methods

### 2.1. Patient Population

We retrospectively recruited patients diagnosed with pit-NET who underwent endoscopic transsphenoidal surgery at Linkou Chang Gung Memorial Hospital, Taiwan, from January 2018 to April 2023. Patients over 18 years old with primary pituitary tumors confirmed by brain MRI were included in our research. Patients with previous sinonasal surgery, head and neck radiotherapy history, under 18 years old, or pregnant were excluded. Further pathological reports that showed metastatic carcinoma, necrosis, chronic inflammation, fibrous dysplasia, and Rathke cleft cysts were also excluded. The enrolled patients completed postoperative medical treatment and regular follow-up at the neurosurgery (NS) department for at least 1 year and had follow-up at the otolaryngology department for at least 3 months. The clinical diagnosis of postoperative sphenoid sinusitis was made by transnasal endoscopic findings or computed tomography (CT) scans in the otolaryngology outpatient department. This study was approved by the Institutional Review Board of Chang Gung Memorial Hospital.

### 2.2. Surgical Treatment and Postoperative Regimen

Endoscopic transsphenoidal pituitary surgery has three phases: the nasal phase, sphenoid phase, and sellar phase [8]. The operation is performed on both nostrils and usually starts on the right nostril, extending up to the anterior sphenoid. The lower half of the right middle turbinate and partial posterior part of the septum were removed to gain enough space for both the endoscope and one instrument inserted through the right nostril, whereas the other instrument was inserted through the left nostril. The procedure became a two-surgeon (one neurosurgeon and one otolaryngologist), two-nostril operation. The pituitary tumor was removed by using an internal debulking procedure. After surgery, all patients were scheduled for regular follow-ups at both outpatient departments. All patients were treated with cortisone acetate 31.25 mg per day for 1 week postoperatively, and cortisol level was checked during the first NS outpatient follow-up. Additional cortisone acetate was prescribed if persistently low cortisol levels were noted. Regular follow-up at the otolaryngology (ENT) outpatient department (OPD) was arranged weekly in the first month postoperatively and monthly to 6 months postoperatively. The bilateral nostril was cleaned at the ENT OPD and nasal discharge and crusting was all removed by suction and forceps. Nasal irrigation was performed daily for at least 6 months by patients themselves after the first local treatment at ENT OPD. Endoscopic examination was performed to check the healing of the surgical wound. Prophylactic antibiotics were taken one week after the surgery. Postoperative sphenoid sinusitis was defined as the occurrence of purulent sinus discharge from the sphenoid ostium or fungal ball under endoscopic examination. If sinusitis was diagnosed by an otolaryngologist, nasal irrigation and postsurgical debridement were performed, and an antibiotic prescription was possibly given. Sinus CT was arranged if the medication failed, and further sinus surgery was indicated. A mild polypoid appearance of the sphenoid sinus mucosa and clear nasal discharge were not considered episodes of sphenoid sinusitis.

### 2.3. Statistical Analysis

Statistical analysis was performed using MedCalc and GraphPad Prism version 5.0. The Mann–Whitney U test was used to compare significant differences between groups. A receiver-operating characteristic (ROC) curve was used to determine the cutoff in the derivation cohort. The best cutoff value was found using Youden’s index. Numerical data were analyzed with the Kruskal–Wallis test between groups and are presented herein as the median with 95% confidence interval (CI). Categorical data were analyzed with the chi-squared test and are presented as percentages. In the multivariate logistic regression (MLR) analysis, logistic regression was used for the evaluation of independent factors for the occurrence of postoperative sinusitis, and coefficients and odds with 95% confidence intervals are presented. A value of *p* < 0.05 was considered to indicate statistical significance. Patients who had or had not developed sphenoid sinusitis were compared using univariate analysis (the chi-squared test was used for nominal variables and Student’s *t* test and Mann–Whitney U test for numerical variables). Variables with a *p* value < 0.1 were put into the subsequent logistic regression to evaluate independent factors for disease recurrence [9]. The performance of the MLR analysis was assessed by determining its discrimination and calibration. The discrimination was measured by calculating the area under the receiver-operating characteristic curve (AUROC). The calibration was assessed using the Hosmer–Lemeshow test Ĉ-test (with *p* > 0.05 indicating no significant difference between the predicted and observed outcomes) [10].

## 3. Results

### 3.1. Study Population

Our analysis included a total of 203 patients who had completed the 1-year postoperative follow-up. Among them, 17 patients were diagnosed with isolated sphenoid sinusitis, with 12 cases attributed to acute rhinosinusitis and 5 cases to fungal sinusitis (as illustrated in Figure 1). The mean age of the sinusitis group was 55.1 years, with males comprising 54.5% of the cohort. Upon comparing patients in the sinusitis group with those in the non-sinusitis groups, no significant differences were observed regarding age, sex distribution, presence of diabetes mellitus (DM), smoking habits, nasoseptal flap preparation, serum white blood cell level, or serum eosinophil count and tumor size. These findings suggest that these demographic and clinical factors may not play a substantial role in predisposing patients to postoperative sinusitis following transsphenoidal endoscopic pituitary surgery.

### 3.2. Risk Factors for Postoperative Sinusitis

The findings from our study unveiled significant associations between postoperative sphenoid sinusitis and various factors, as highlighted in the univariate analysis (Table 1). Specifically, pituitary tumor classification (nonfunctioning pit-NET or functioning pit-NET), preoperative cortisol level, and postoperative cortisol level emerged as influential factors. Subsequent multivariate logistic regression (MLR) analysis delved more deeply, identifying nonfunctioning pituitary adenoma (odds: 3.71, 95% CI: 1.09–12.58, *p* = 0.036) and postoperative serum cortisol level (odds: 0.82, 95% CI: 0.69–0.99, *p* = 0.035) as independent factors associated with postoperative sphenoid sinusitis (Table 2). The MLR model shed light on the specific dynamics, revealing that patients with nonfunctioning pituitary adenoma had 3.71 times the likelihood of developing postoperative sinusitis. Additionally, for each increment in postoperative cortisol level following transsphenoidal surgery, the odds of developing sphenoid sinusitis decreased by 18%. Importantly, the MLR model exhibited robust discriminatory power with an AUROC of 0.823 and satisfactory calibration (Hosmer–Lemeshow Ĉ test, *p* = 0.0625) (Figure 2).

### 3.3. Case Illustration

Table 3 provides a comprehensive overview of 17 patients who were diagnosed with postoperative isolated sphenoid sinusitis. Among these individuals, the average age was found to be 55.1 years, with a nearly equal distribution between males and females, with nine males and eight females. Patients were further categorized into two groups based on the presence of pituitary neuroendocrine tumors (pit-NETs), with nine patients having nonfunctioning pit-NETs and eight patients having functioning pit-NETs.

Throughout the postoperative follow-up period, thirteen patients reported experiencing various symptoms indicative of sinusitis, including purulent rhinorrhea, foul odor, headache, nasal obstruction, sputum with and without blood, periorbital pain, hyposmia, and cough. Nasal endoscopic examinations unveiled prevalent findings, such as mucopus accumulation over the sphenoid ostium, presence of fungal balls, and hyphae accompanied by cheesy white material within the sphenoid sinus ostium. Notably, two patients could not undergo endoscopy due to restrictions imposed by the COVID-19 pandemic, leading to a diagnosis based solely on clinical symptoms and medical history.

Typical CT and endoscopic findings of isolated sphenoid fungal sinusitis are illustrated in Figure 3. CT scans showcased bilateral sphenoid sinus opacification without extension to adjacent sinuses, along with focal hyperdense areas indicative of calcium phosphate and calcium deposits. Among the patient cohort, five individuals were conclusively diagnosed with fungal sinusitis based on either CT or endoscopic findings.

All patients developed sphenoid sinusitis within three months postsurgery. While medical treatment proved effective for fifteen patients, two individuals failed to respond, necessitating further endoscopic sinus surgery. Moreover, comorbidities were noted within the patient population, with four individuals having DM and one being a habitual smoker. These factors could potentially contribute to the complexity of their sinusitis management and treatment outcomes.

## 4. Discussion

In our comprehensive study, we meticulously examined a cohort of 17 patients exclusively from a singular tertiary center, all of whom presented with isolated sphenoid sinusitis subsequent to undergoing transsphenoidal endoscopic pituitary surgery. Notably, our investigation unveiled a significant association between the classification of pit-NETs and postoperative cortisol levels with the occurrence of isolated sphenoid sinusitis postsurgery. What distinguishes our research is its unique contribution to the existing body of knowledge. Remarkably, there exists no prior study within the English literature to have delved into the analysis of risk factors specifically tailored to this clinical scenario. This distinct focus enhances the relevance and applicability of our findings, offering valuable insights into the intricate interplay between surgical interventions, tumor characteristics, and postoperative complications, particularly in the context of sphenoid sinusitis following transsphenoidal pituitary surgery.

Pit-NETs, formerly called pituitary adenomas, are the most common type of sella turcica tumor and are of benign nature [11]. Their prevalence ranges between 78 and 94 cases per 100,000 habitants based on recent studies [12,13]. Nonfunctioning pit-NETs, characterized by the absence of clinical and biological hormonal secretion evidence, constitute a substantial proportion, accounting for 25–40% of all pituitary Pit-NETs [14,15,16], with prolactinomas taking the lead with a prevalence of 40–55%. The remaining subtypes occur less frequently, including GH-secreting adenomas in 10% of cases, ACTH-secreting adenomas in 1–5% of cases, and TSH-secreting adenomas in less than 1% of cases [17].

Common complications of transsphenoidal endoscopic surgery include CSF leakage, meningitis, diabetes insipidus, electrolyte imbalance, neurological deterioration, and vascular injury, as documented in previous literature [18,19,20]. However, postoperative isolated sphenoid sinusitis complications have received scant attention in prior studies. Nonetheless, our attention was piqued by encounters with several cases during outpatient follow-up, prompting a deeper investigation into this clinical phenomenon.

Cheng et al. [21] conducted a comprehensive analysis encompassing a total of 129 cases to explore nasal complications following transsphenoidal resection of pituitary neoplasms. Their findings revealed a notable incidence rate of postoperative nasal complications, reaching as high as 20.1%. Among these complications, nasal hemorrhage accounted for 4.8%, cerebrospinal fluid rhinorrhea for 6.9%, sphenoid sinusitis for 2.3%, atrophic rhinitis for 1.6%, olfactory disorder for 1.6%, nasal septum perforation for 0.8%, and nasal adhesion for 2.3%. Notably, three cases of postoperative sphenoid sinusitis were effectively managed through a regimen of regular nasal endoscopic cavity cleaning, intranasal corticosteroids, and nasal irrigation.

Sinusitis is a condition characterized by inflammation of the mucous membranes in the nasal cavity and paranasal sinuses, leading to the accumulation of fluid within these cavities. It presents in different forms: acute or subacute if symptoms persist for three months or less, and chronic sinusitis if they extend beyond three months [22]. Specifically, when inflammation affects the sphenoid sinus, it is termed sphenoid sinusitis. This condition can either be confined solely to the sphenoid sinus or may affect multiple sinuses concurrently. Sphenoid sinusitis can present with various symptoms, such as headaches, facial pain, nasal congestion, and postnasal drip. Diagnosis typically involves a combination of medical history assessment, physical examination, and imaging studies like CT scans or MRI. Treatment approaches may vary depending on the severity and underlying cause, ranging from conservative measures such as nasal irrigation and decongestants to antibiotics or even surgery in cases of chronic or recurrent sinusitis that do not respond to other treatments. Early diagnosis and appropriate management are crucial to prevent complications and improve quality of life for individuals affected by sphenoid sinusitis.

Isolated sphenoid sinusitis is a relatively rare clinical condition, representing 1–2% of sinus infections [22], yet it tends to cause more severe complications, such as intracranial invasion and cavernous sinus thrombosis [23,24], and the rate of cranial neuropathy is reported to be 16.3% [25]. Its treatment requires medical intervention, nasal irrigation, and sometimes surgical intervention. The pathophysiology of sphenoid sinusitis involves sphenoid ostium blockage, impaired mucociliary clearance, stasis, and secondary infections, including those caused by bacteria and fungal hyphae [6,7]. The most frequent symptoms of sphenoid sinusitis include retro-orbital headache and visual symptoms such as blurred vision, diplopia, transient visual loss, and isolated oculomotor palsy, and nasal obstruction and epistaxis [26,27]. Highly inflammatory diseases like fungal infections may increase the risk of reossification of the sphenoid ostium following sphenoidotomy [22].

Reviewing published studies in the English literature, Lu et al. [7] reported 20 cases of isolated sphenoid sinusitis or mucocele as a rare complication of endonasal transsphenoidal pituitary surgery in 2009, with 6.2% incidence. Patients who underwent a small sphenoidotomy had a higher potential of developing sinusitis than patients who received wider sphenoidotomy. Also, patients under more frequent and longer duration of postoperative nasal care by an otolaryngologist showed a lower likelihood of developing postoperative sinusitis. Batra et al. [6] reported 15 of 200 patients developed sinusitis after transsphenoidal surgery, with the most common symptoms included headaches and nasal discharge. Five patients required further endoscopic sphenoidotomy, fungal balls were identified in three of the patients, mucocele formation was encountered in one case, and one presented with infected fat graft. Chung et al. [28] reported a 56-year-old female patient diagnosed with isolated fungal sphenoid sinusitis 10 years after a transsphenoidal approach for pit-NETs and a history of skull base repair 5 years prior to the event. She presented with symptoms of postnasal drip and foul odor, and endoscopic examination revealed purulent discharge and swollen mucosa of sphenoid sinus. Sajko et al. [29] reported seven patients with sphenoid sinus aspergilloma after transsphenoidal surgery for pit-NETs in Croatia in a 10-year period. Kim et al. [30] presented a case of a fungal ball in the sphenoid mucocele with transsphenoidal surgery over 20 years ago. However, there is currently no large-scale etiological analysis of postoperative sphenoid sinusitis in the previous literature.

In the current study, all 17 patients developed postoperative sphenoid sinusitis within three months after surgery. Hence, we highly suspect that this clinical condition is related to the procedure of transsphenoidal endoscopic pituitary surgery. Our study revealed that postoperative sinusitis is related to the classification of pit-NETs and postoperative cortisol levels during the first outpatient follow-up. Firstly, a transsphenoidal endoscopic operation for a nonfunctioning pit-NET was more likely to lead to the development of postoperative sinusitis than an operation for a functioning pit-NET. Nonfunctioning pit-NETs are typically initially asymptomatic and are usually detected when they grow larger, causing mass effects on surrounding structures [31]. This growth process might result in more significant chronic inflammation or alterations in the anatomy structure within the sinus cavity even before surgical intervention. Such preexisting conditions could potentially predispose individuals to an increased risk of postoperative sinusitis. Secondly, we found that the lower the postoperative cortisol level, the higher the chance of postoperative sinusitis. The hypothesis is that a longer usage of postoperative cortisone acetate was prescribed when a lower cortisol level was noted. Prolonged usage of cortisone acetate may cause an immunocompromised status and lead to higher rates of developing sphenoid sinusitis.

The limitations of our study warrant careful consideration. Firstly, our study population was drawn exclusively from a single institution—a tertiary referral center in northern Taiwan. This factor may introduce selection bias, particularly concerning the severity of the disease and the geographical distribution of patients. Furthermore, the relatively small number of observed complications and the retrospective design of the study are notable limitations. Also, the rate of isolated sphenoid sinusitis in our cohort was 8.3%, a rather high incidence comparing to previous retrospective studies reported. Our theory was that all our patients were regularly assigned to our ENT department for postoperative follow-up. Endoscopy examination was performed for evaluating the presence of sphenoid sinusitis. However, 4 of our 17 reported sphenoid sinusitis patients did not present any nasal symptoms, which may be omitted if no regular follow-up were assigned, which may in itself result in a higher prevalence of postoperative sphenoid rate than that reported. Specifically, employing a larger and more diverse study population would enhance the generalizability of the findings. Additionally, extending the follow-up period would allow for a more comprehensive assessment of the outcomes. By addressing these limitations, future studies can provide more robust evidence and contribute to a deeper understanding of the topic at hand.

In conclusion, our findings underscore the significance of nonfunctioning pit-NETs and reduced postoperative cortisol levels as potential risk factors for postoperative sinusitis following transsphenoidal endoscopic pituitary surgery. By identifying these risk factors, health-care providers can better tailor postoperative care strategies, potentially reducing the incidence and severity of sinusitis complications in patients undergoing this surgical intervention. Such knowledge may lead to improved patient outcomes and enhanced quality of care within the realm of pituitary tumor management.

## Figures and Tables

**Figure 1 diagnostics-14-00758-f001:**
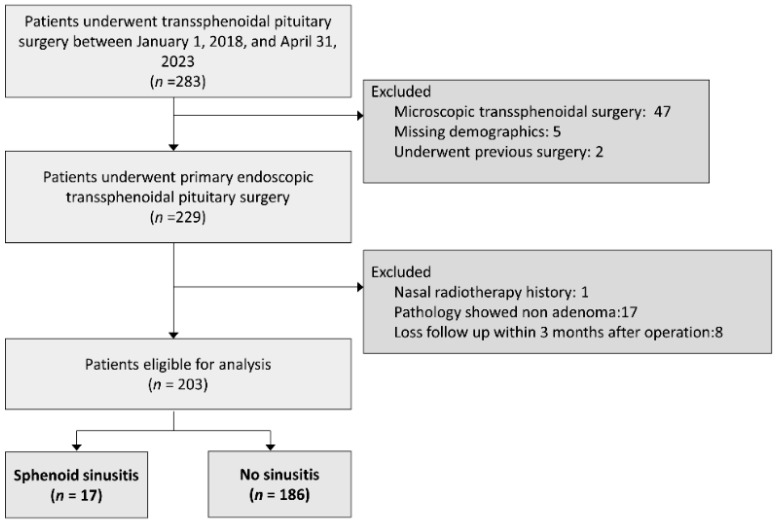
Patient flow diagram.

**Figure 2 diagnostics-14-00758-f002:**
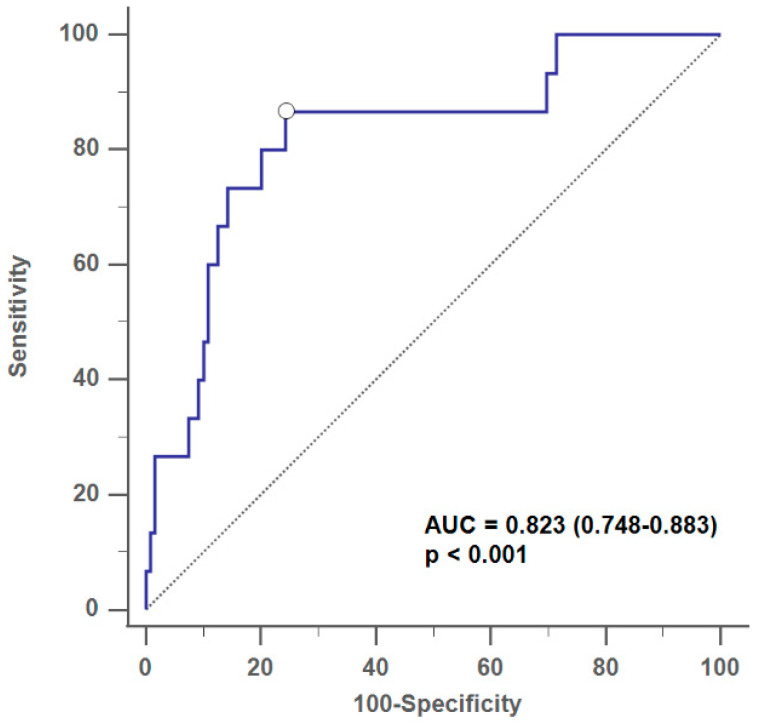
Receiver-operating characteristic (ROC) curve showing the discriminant performance of the multivariate logistic regression analysis with AUC= 0.823 for the evaluation of risk of developing postoperative isolated sphenoid sinusitis (*p* < 0.001).

**Figure 3 diagnostics-14-00758-f003:**
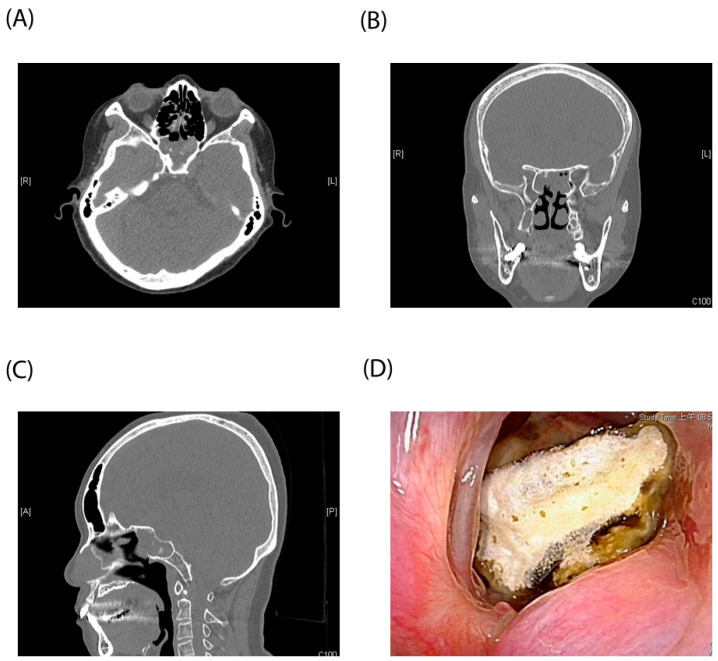
Typical CT and endoscopic findings of isolated sphenoid fungal sinusitis. (**A**) axial view (**B**) coronal view (**C**) sagittal view of Computed tomographyo (**D**) Nasoendoscopy finding of hyphae accompanied by cheesy white material within the sphenoid sinus ostium.

**Table 1 diagnostics-14-00758-t001:** Comparisons between isolated sphenoid sinusitis group and non-sinusitis group.

Variables	Sinusitis (*n* = 17)	No Sinusitis (*n* = 186)	*p* Value
Male	6	97	0.184
Age	55.1 (41–69.2)	53.5 (39.7–67.3)	0.649
DM	4 (23.5)	32 (17.2)	0.514
Smoking	0 (0)	1 (5.3)	1.000
Nonfunctioning pituitary adenoma	9 (52.9)	53 (28.5)	0.037 *
Nasoseptal flap preparation	4 (23.5)	54 (29.0)	0.343
Preoperative cortisol level	5.4 (1.6–9.4)	8.7 (7.4–9.7)	0.024 *
Postoperative cortisol level	4.3 (2.1–7.3)	7.4 (6.8–8.4)	0.014 *
WBC (×10^3^)	7.4 (6.8–8.5)	6.6 (6.3–6.9)	0.223
Eosinophil count	110.5 (95.2–177.9)	122 (99.7–136.3)	0.611
Tumor size (longest diameter, mm)	14.8 (12.6–20.2)	15.3 (13.9–24.3)	0.462

Numerical data: median (95% CI). Categorial data: *n* (%). * *p* < 0.05. Abbreviations: DM, diabetes mellitus.

**Table 2 diagnostics-14-00758-t002:** Subsequent multivariate logistic regression analysis for the risk of developing postoperative isolated sphenoid sinusitis.

Variables	Coefficient	Odds Ratio	95% CI	*p* Value
Nonfunctioning pituitary adenoma	1.310	3.71	1.09–12.58	0.036 *
Postoperative cortisol level	−0.194	0.82	0.69–0.99	0.035 *
Preoperative cortisol level	-	-		0.062

* *p* < 0.05.

**Table 3 diagnostics-14-00758-t003:** Isolated sphenoid sinusitis cases illustration.

	Age/Gender	Pit-NET Type	Nasal Symptoms	Endoscopic Findings	Follow-Up Duration to Sinusitis (M)	Operation for Sinusitis	DM	Smoking
Case 1	37/F	Nonfunctioning pit-NET	Foul odor and clear rhinorrhea	Mucopus over sphenoid sinus ostium	1	N	1	0
Case 2	63/F	Nonfunctioning pit-NET	None	Mucopus over sphenoid sinus ostium	1	N	1	0
Case 3	49/F	Nonfunctioning pit-NET	Purulent rhinorrhea	Mucopus over sphenoid sinus ostium	3	N	0	0
Case 4	58/M	Nonfunctioning pit-NET	Headache	Mucopus over sphenoid sinus ostium	1	N	0	0
Case 5	77/F	Functioning pit-NET	Right nasal obstruction	Mucopus over sphenoid sinus ostium	1	N	0	0
Case 6	39/F	Nonfunctioning pit-NET	Purulent rhinorrhea and post nasal dripping	Mucopus over sphenoid sinus ostium	1	N	0	0
Case 7	76/F	Nonfunctioning pit-NET	Blood-tinged sputum	Fungal ball over sphenoid sinus ostium	2	Y	0	0
Case 8	29/M	Functioning pit-NET	None	Fungal ball over sphenoid sinus ostium	2	N	0	0
Case 9	49/F	Functioning pit-NET	Nasal obstruction	NA (due to COVID pandemic)	0.5	N	1	0
Case 10	40/M	Nonfunctioning pit-NET	Purulent rhinorrhea	Polypoid change with mucopus over sphenoid sinus ostium	1	N	0	0
Case 11	74/F	Functioning pit-NET	None	Mucopus over sphenoid sinus ostium	0.5	Y	0	0
Case 12	54/M	Functioning pit-NET	Blood-tinged sputum	Fungal hyphae over sphenoid ostium	1	N	0	0
Case 13	68/F	Functioning pit-NET	None	Mucopus over sphenoid sinus ostium	0.5	N	0	0
Case 14	58/M	Nonfunctioning pit-NET	Foul odor	Stenosis of sphenoidotomy with mucopus	2	N	0	0
Case 15	53/F	Functioning pit-NET	Right periorbital pain and hyposmia	Right synechia and mucopus over sphenoid ostium	1	N	0	0
Case 16	50/F	Nonfunctioning pit-NET	Sputum and cough	Mucopus over sphenoid sinus ostium	3	N	1	0
Case 17	63/M	Functioning odd pit-NET	Purulent rhinorrhea	NA (due to COVID pandemic)	1.5	N	0	0

Abbreviations: pit-NET, pituitary neuroendocrine tumor; DM, diabetes mellitus; M, male; F, female; Y, yes; N, no, NA, not available.

## Data Availability

The datasets generated during and/or analyzed during the current study are available from the corresponding author on reasonable request.

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
