# Peer review of "Risk Factors for Isolated Sphenoid Sinusitis after Endoscopic Endonasal Transsphenoidal Pituitary Surgery"

_diagnostics, 2024, doi:10.3390/diagnostics14070758_

Round 1

Reviewer 1 Report

Comments and Suggestions for Authors

Dear Authors,

I have carefully reviewed the manuscript titled "Risk factors for isolated sphenoid sinusitis after endoscopic endonasal transsphenoidal pituitary surgery” submitted for consideration in Diagnostics.

After thorough consideration, I regret to inform you that I do not believe the manuscript is suitable for publication in its current form. I have outlined my reasons for this decision below:

1. Although the aim is interesting, the methodology should be extensively revised.

2. It is unusual that nonfunctional adenoma and low cortisol levels postoperatively are connected to sphenoid sinusitis. The authors should exclude the influence of tumour size on sphenoid sinusitis (larger tumours often result in hypopituitarisms). It can not be excluded that this is an accidental connection.

3. Authors should analyse the connection between Cushing's disease and sphenoid sinusitis.

4. There is rather a high incidence of sphenoid sinusitis in this study; is there an explanation?

Reviewer 2 Report

Comments and Suggestions for Authors

Authors evaluate the relevance of sinusitis after transsphenoidal surgery. It is surely an important and not well examined complications after pituitary surgery and I thank authors for this examination. I still have some question for the authors.

It was described that ½ of middle turbinate was removed during the preparation? Was this really performed in all surgeries, since normally, in the case of typical pituitary adenomas, it is not necessary to do so. Could you comment on that and on potential influence on the sinusitis? 

According to the results, if I understand it correctly, more functional than non-functional pituitary adenomas were treated according to table 1(62 patients with non-functioning adenomas) Could you provide the additional data, which kind of tumors were in the cohort? How was the management in Cushing patients? Was there sinusitis relevant?

I do not exactly understand the substitution regiment after surgery. In all patients corticotrope axis was substituted after surgery, but it is not clear how did you examined the potential insufficiency? Did you perform functional testing and what was the time schedule of that? If all your patients got prophylactically cortison after surgery, there could be a bias which should be discussed. 

Do you have a data on quality of life in the patients with sinusitis and comparison with the rest of the cohort? Was one treatment session in the case of sinusitis sufficient or did patients undergo more treatments?

How was the normal nose care of the patients after the surgery?

Reviewer 3 Report

Comments and Suggestions for Authors

The paper deals about occurrence of an underestimate complication of pituitary surgery. This aspect is non so well focused in the literature. The study is retrospective and this represent a limits but the data analysis allows some interesting conclusion. So the study may be considered for the publication.

3.2 paragraph needs to be removed representing a repetion. In the paragraph  3.4 also a repetition is present and needs to be removed (see attached file). Table 3 with 17 patients findings is not so necessary and in my opinion may be removed. I finally suggest a little revision of discussion that should be a little shortened.

Comments on the Quality of English Language

English language is clear and understandable. Only minore revision may be necessary

Round 2

Reviewer 1 Report

Comments and Suggestions for Authors

Dear authors,

Thank you for the comments and improvement of article.